# Prognostic Significance and Emerging Predictive Potential of Interleukin-1β Expression in Oncogene-Driven NSCLC

**DOI:** 10.3390/cancers17172895

**Published:** 2025-09-03

**Authors:** Mengni Guo, Won Jin Jeon, Bowon Joung, Derek Tai, Alexander Gavralidis, Andrew Elliott, Yasmine Baca, David de Semir, Stephen V. Liu, Mark Reeves, Saied Mirshahidi, Hamid Mirshahidi

**Affiliations:** 1Division of Hematology and Oncology, Loma Linda University, Loma Linda, CA 92354, USA; mengniguomd@gmail.com (M.G.); wjjeon@llu.edu (W.J.J.); bjoung@llu.edu (B.J.); agavralidis@llu.edu (A.G.); mereeves@llu.edu (M.R.); 2Department of Internal Medicine, Loma Linda University, Loma Linda, CA 92354, USA; dtai@llu.edu; 3Caris Life Sciences Inc., Phoenix, AZ 85040, USA; aelliott@carisls.com (A.E.); ybaca@carisls.com (Y.B.); ddesemir@carisls.com (D.d.S.); 4Lombardi Comprehensive Cancer Center, Georgetown University, Washington, DC 20007, USA; stephen.liu@gunet.georgetown.edu; 5Loma Linda University Cancer Center, Loma Linda University, Loma Linda, CA 92354, USA; smirshahidi@llu.edu; 6Biospecimen Laboratory, Department of Medicine and Basic Sciences, Loma Linda University, Loma Linda, CA 92354, USA

**Keywords:** interleukin-1β, NSCLC, oncogenic mutations, EGFR, ALK, KRAS, survival outcomes, tumor microenvironment

## Abstract

Lung cancer remains a leading cause of cancer-related death, and new approaches are needed to help understand prognosis and guide treatment. This study examined whether levels of a pro-inflammatory molecule called interleukin-1 beta (IL-1β) in tumors are associated with survival in non-small cell lung cancer (NSCLC). Over 21,000 tumors were analyzed, linking IL-1β expression levels with real-world survival data. In certain genetic subtypes of NSCLC, lower IL-1β expression was associated with longer survival. However, in tumors without these mutations, IL-1β levels appeared to have less impact. These findings suggest that IL-1β may play a role in the biology of some lung cancers and support future research into its potential as a biomarker or treatment target.

## 1. Introduction

Non-small cell lung cancer (NSCLC) accounts for 80–85% of all lung cancer cases in the United States [1] and is the most prevalent histology of lung cancer globally. NSCLC is often diagnosed at an advanced stage due to its asymptomatic nature in early stages, leading to a poor prognosis. As a result, NSCLC carries the highest mortality rate among all cancers worldwide [2].

Recent breakthroughs in understanding the molecular mechanisms underlying NSCLC have enabled the development of targeted therapies, including those directed at epidermal growth factor receptor (EGFR) mutations and anaplastic lymphoma kinase (ALK) rearrangements, and the incorporation of immune-checkpoint inhibitors (ICI). These therapeutic advancements have markedly enhanced outcomes for patients with advanced NSCLC, making personalized treatment approaches possible. Nonetheless, the overall five-year survival rate remains low, with the mortality rate of stage IV NSCLC often exceeding 90% within five years of diagnosis [3].

These statistics underscore the critical need to identify novel therapeutic strategies and biomarkers that can guide prognosis and treatment response. Among these biomarkers, interleukin-1β (IL-1β), a protumor cytokine [4,5,6,7,8,9,10], stands out as a potential prognostic and predictive marker. IL-1β is secreted primarily by activated monocytes, macrophages, and neutrophils, though tumor and stromal cells can also produce it [11,12,13,14]. Within the tumor microenvironment, IL-1β acts on endothelial cells, fibroblasts, and immune subsets to promote angiogenesis, recruit immunosuppressive myeloid cells, and sustain chronic inflammation that facilitates tumor progression [4,12,15,16,17]. Beyond NSCLC, IL-1β has been implicated in breast, gastric, and colorectal cancers, where it contributes to tumor growth, metastasis, and therapy resistance [6,12,18,19]. These broader roles highlight IL-1β as a common driver of cancer progression and a potential biomarker and therapeutic target. IL-1β activates a cascade of inflammatory mediators, fostering a protumoral TME and facilitating tumor progression. Elevated levels of IL-1β in the peripheral blood and tumor tissues have been linked to poorer survival rates in NSCLC, indicating the importance of further exploration into the prognostic implications of IL-1β [20,21,22]. Moreover, IL-1β expression has been associated with resistance to EGFR-targeted tyrosine kinase inhibitors (TKIs), impacting treatment responses in NSCLC patients with actionable mutations [23]. The CANOPY studies [24,25,26] failed to demonstrate survival benefit when IL-1β inhibition was combined with chemoimmunotherapy in patients lacking oncogenic driver mutations, pointing to gaps in understanding the prognostic and predictive significance of IL-1β, especially in the context of oncogenic drivers and ICI therapy.

In this study, we present the results of a large retrospective database analysis examining the relationship between tumor IL-1β expression and survival outcomes in NSCLC patients, particularly those with actionable oncogenic mutations.

## 2. Materials and Methods

### 2.1. Patient Samples

A total of 21,698 formalin-fixed, paraffin-embedded (FFPE) tumor samples from patients with NSCLC were submitted to a CLIA-certified commercial laboratory (Caris Life Sciences, Phoenix, AZ, USA) for molecular profiling. Samples underwent next-generation sequencing (NGS) of DNA (592-gene panel or whole exome sequencing) and RNA (whole transcriptome sequencing, WTS), and immunohistochemistry (IHC) for programmed death-ligand 1 (PD-L1). Tumors were then stratified by IL-1β expression quartiles (Q1: lowest 25% expression; Q4: highest 25% expression) based on the distribution of expression values in the full cohort. A median (50%) cut-off was also evaluated but did not reveal significant differences in survival outcomes; therefore, quartile stratification was used for the primary analyses. Real-world survival outcomes were obtained from insurance claims and calculated from either time of tissue collection or start of treatment to last contact or time on treatment (TOT).

### 2.2. DNA NGS

Tumor enrichment was performed via manual microdissection of FFPE tissue samples. Genomic DNA was extracted and subjected to NGS using the NextSeq or NovaSeq 6000 Platforms (Illumina, Inc., San Diego, CA, USA). A custom SureSelect XT assay (Agilent Technologies, Santa Clara, CA, USA) was used to enrich exonic regions of 592 genes. Samples sequenced on the NovaSeq 6000 platform included over 700 clinically relevant genes. All variants were detected with a confidence level exceeding 99% based on allele frequency and amplicon coverage, with an average sequencing depth of over 500× and an analytic sensitivity threshold of 5%. Variants were reviewed and classified by board-certified molecular geneticists per American College of Medical Genetics and Genomics (ACMG) guidelines.

### 2.3. Tumor Mutational Burden (TMB) and MSI-H

TMB was measured by counting all non-synonymous missense, nonsense, in-frame insertion/deletion and frameshift mutations found per tumor, which had not been previously reported as germline alterations in dbSNP151 in the Genome Aggregation Database (gnomAD) or deemed benign variants by Caris geneticists. A cutoff of ≥10 mutations (mt)/MB was used based on the KEYNOTE-158 trial, which showed higher response rates in tumors with a TMB of ≥10 mt/MB [27].

Microsatellite stability (MSI) or mismatch repair (MMR) status was determined using fragment analysis (Promega Corporation, Madison, WI, USA), IHC [MLH1, M1 antibody; MSH2, G2191129 antibody; MSH6, 44 antibody; and PMS2, EPR3947 antibody], and NGS. For tumors tested with the NextSeq platform, 7000 microsatellite loci were compared with the hg19 reference genome from the University of California, Berkeley, CA.

### 2.4. RNA Expression (WTS)

RNA was extracted from FFPE tumor samples and sequenced using the Illumina NovaSeq platform (Illumina, Inc., San Diego, CA, USA) and Agilent SureSelect Human All Exon V7 bait panel (Agilent Technologies, Santa Clara, CA, USA). Expression levels were reported as transcripts per million (TPM). Immune cell fractions in the TME were estimated using the quanTIseq deconvolution algorithm, which infers immune composition from bulk RNA-seq data based on validated cell-type-specific transcriptomic signatures [28].

### 2.5. IHC

IHC was performed on FFPE slides using automated staining techniques per the manufacturer’s instructions and were optimized and validated per CLIA/CAO and ISO requirements. A board-certified pathologist independently reviewed all IHC results. The primary PD-L1 antibody clone was 22c3 (Dako, Glostrup, Denmark). Tumor proportion score (TPS) was defined as the percentage of viable tumor cells showing partial or complete membrane staining at any intensity. TPS ≥ 1% was considered positive; TPS ≥ 50% defined high PD-L1 expression.

### 2.6. Real-World Clinical Outcomes

Overall survival (OS) and TOT data were derived from insurance claims. OS was calculated from the time of tissue collection (proxy for diagnosis) or from the start of treatment to the last point of contact/death. TOT was calculated from the first to last treatment date. Treatments included platinum-based chemotherapy and pembrolizumab.

### 2.7. Statistical Analysis

Chi-square, Fisher’s exact, and Mann–Whitney U tests were used for comparisons, with *p* values being adjusted for multiple comparisons (*q* < 0.05). Survival outcomes (OS and TOT) were assessed using Cox proportional hazards models to calculate hazard ratios (HRs) and log-rank tests for *p* values.

### 2.8. Compliance Statement

This retrospective study was conducted under Caris Life Sciences’ Research Data Banking protocol, which was reviewed and granted IRB exemption by the WCG IRB. The study adhered to the ethical guidelines of the Declaration of Helsinki, the Belmont Report, and the U.S. Common Rule.

## 3. Results

### 3.1. Study Population and Mutational Status

The study included 21,698 NSCLC samples (Table 1). Median age was similar across groups. Adenocarcinomas were more frequent in Q1 (lowest IL-1β expression) than Q4 (72.2% vs. 44.3%, *p* < 0.001), while squamous cell carcinomas (SCCs) were more common in Q4 (12.1% vs. 36.4%, *p* < 0.001). Of the total samples, 64.4% (*n* = 13,974) were derived from primary tumor sites, 32.9% (*n* = 7137) from metastatic lesions, and 2.7% (*n* = 587) had unknown origin. Staging information was unavailable.

### 3.2. IL-1β Expression and TME Characteristics

To explore the relationship between IL-1β expression and the TME in NSCLC, we compared immune cell infiltration between Q1 and Q4 using quanTIseq (original R package [28]). Our analysis demonstrated that neutrophils, M1 macrophages, regulatory T-cells, CD8+ T cells and myeloid dendritic cells were significantly enriched in Q4 compared to Q1 (*p* < 0.001, Table 2).

### 3.3. IL-1β Expression Variations in Different Histologic Subgroups and Oncogenic Mutations

IL-1β expression varied across histologic and molecular subtypes (Table 3). In tumors without EGFR, KRAS, or ALK alterations, adenocarcinomas showed lower IL-1β expression (median: 4.92 TPM) than SCC (median: 8.47 TPM) (*p* < 0.001). A similar pattern was observed in EGFR-mutant (median: 4.44 vs. 9.98 TPM) (*p* < 0.001) and KRAS-mutant (median: 4.78 vs. 10.14 TPM) (*p* < 0.001) tumors. In ALK fusion-positive tumors, adenocarcinoma showed slightly higher IL-1β expression (median: 5.16 vs. 4.88 TPM), but this difference was not statistically significant.

### 3.4. Associations Between IL-1β Expression and Genetic Alterations in NSCLC

IL-1β expression was associated with key genetic alterations in NSCLC (Figure 1). TP53 mutations were more common in Q4 (76.0%) vs. Q1 (57.3%). High TMB and elevated PD-L1 expression were more enriched in Q4 (44.0% and 66.6%, respectively) compared to Q1 (37.3% and 43.9%, respectively). In contrast, EGFR, KRAS, BRAF, STK11, and KEAP1 mutations were more prevalent in Q1.

### 3.5. Association of IL-1β Expression with Survival Outcomes in NSCLC

In the entire NSCLC cohort, lower IL-1β expression was associated with modestly longer OS compared to higher expression levels. Specifically, the median OS was 19.5 months in Q1 compared to 17.4 months in Q4, with a HR of 0.94 (95% CI: 0.91–0.97, *p* < 0.0001) (Figure 2a).

#### 3.5.1. Impact of IL-1β Expression in NSCLC Without Actionable Oncogenic Mutations

In NSCLC without actionable oncogenic mutations, IL-1β had no impact on OS (HR 0.98, 95% CI: 0.93–1.04, *p* = 0.48) (Appendix A). In adenocarcinomas without oncogenic mutations, high IL-1β was linked to improved OS (15.4 months in Q1 vs. 19.0 months in Q4; HR 1.15, 95% CI: 1.05–1.26, *p* = 0.003) (Appendix A). However, in SCC, high IL-1β was associated with worse OS (18.1 vs. 14.2 months; HR 0.86, 95% CI: 0.78–0.93, *p* < 0.001) (Appendix A).

#### 3.5.2. Impact of IL-1β Expression in EGFR-Mutated NSCLC

A total of 1663 EGFR-mutated NSCLC cases, including 1454 adenocarcinoma and 33 squamous histology, were analyzed. High IL-1β expression correlated with worse OS (33.3 months in Q1 vs. 26.8 months in Q4; HR 0.81, 95% CI: 0.71–0.92, *p* = 0.001) (Figure 2b). This OS advantage was also observed in EGFR-mutated adenocarcinomas (36.7 vs. 27.2 months; HR 0.76, 95% CI: 0.66–0.87, *p* < 0.001) (Figure 2c). However, the difference in OS among SCC patients was not statistically significant (Figure 2d).

#### 3.5.3. Impact of IL-1β Expression in NSCLC with ALK Fusions

In NSCLC patients with ALK fusions, IL-1β expression significantly impacted OS. Patients with low IL-1β expression had improved OS than those with high expression (53.0 vs. 35.2 months, HR 0.62, 95% CI: 0.45–0.84, *p* = 0.002) (Figure 2e). ALK fusion-positive adenocarcinoma showed a similar pattern (49.1 vs. 40.3 months; HR 0.69, 95% CI: 0.49 –0.97, *p* = 0.034) (Figure 2f). Due to the small number of SCC cases with ALK fusions, further analysis within this histological subtype could not be conducted.

#### 3.5.4. Impact of IL-1β Expression in KRAS-Mutated NSCLC

In NSCLC patients harboring KRAS mutations, IL-1β expression did not significantly affect OS in the overall population, adenocarcinoma or SCC subgroups. For the entire KRAS-mutant cohort, the median OS was 19.1 months in Q1 and 17.0 months in Q4 (HR 0.96, 95% CI: 0.89–1.03, *p* = 0.24) (Appendix A). In KRAS-mutant adenocarcinoma, the median OS was 20.9 months in Q1 compared to 20.2 months in Q4 (HR 0.99, 95% CI: 0.91–1.08, *p* = 0.75) (Appendix A). In the KRAS-mutant SCC subgroup, median OS was 7.9 months in Q1 compared to 9.1 months in Q4 (HR 1.31, 95% CI: 0.88–1.95, *p* = 0.19) (Appendix A).

#### 3.5.5. Impact of IL-1β Expression on Survival Outcomes in Patients Treated with Immunotherapy

In patients with NSCLC treated with pembrolizumab, IL-1β expression did not significantly impact OS (HR 1.03, 95% CI: 0.97–1.09, *p* = 0.37) (Appendix A). For TOT, Q4 showed a statistically significant but not clinically meaningful survival benefit over Q1 (5.8 vs. 6.0 months; HR 1.06, 95% CI: 1.01–1.13, *p* = 0.029) (Appendix A). In KRAS-mutant adenocarcinoma subgroup, OS was similar between Q1 and Q4 groups (HR 1.14, 95% CI 0.98–1.32, *p* = 0.10) (Appendix A). High IL-1β expressors demonstrated better TOT compared to low expressors, with a median TOT of 7.4 months for Q4 and 6.4 months for Q1 (HR 1.15, 95% CI: 1.01–1.31, *p* = 0.04) (Appendix A). Due to the limited sample size, no conclusions could be drawn for KRAS-mutant SCC.

## 4. Discussion

In this study, we demonstrated that high IL-1β expression is associated with worse survival in patients with NSCLC harboring oncogenic driver mutations, particularly EGFR and ALK alterations. This finding aligns with previous literature, which linked elevated IL-1β in serum and tumor tissue to poor prognosis [4,29]. In contrast, among patients without actionable driver mutations, IL-1β levels did not significantly influence survival. These findings highlight the complex role of IL-1β in NSCLC, particularly its interplay with oncogenic drivers and TME, and suggest a potential therapeutic opportunity for targeting IL-1β in molecularly defined subgroups.

Chronic inflammation is a well-established contributor to cancer progression through tumorigenesis, angiogenesis, metastasis, and immune modulation [17,30,31,32]. The incidental finding from the CANTOS trial, where IL-1β inhibition with canakinumab reduced lung cancer incidence and mortality, prompted the CANOPY trials, which evaluated IL-1β blockade in NSCLC [24,26,33,34,35]. However, the CANOPY trials did not demonstrate a survival advantage when canakinumab was added to chemotherapy or immunotherapy, and in some cases, the combination therapy increased the risk of severe infections. Our findings are consistent with these results, as IL-1β expression did not affect OS in NSCLC subgroup without targetable oncogenic mutations. This suggests that in these tumors, alternative pathways may drive survival and proliferation, rendering IL-1β less critical. These observations also align with the broader understanding that tumors without targetable oncogenic mutations tend to be more genetically and biologically heterogeneous, complicating the identification of universal biomarkers and therapeutic targets.

Conversely, among tumors with driver mutations, IL-1β emerged as a strong prognostic marker. In EGFR-mutant NSCLC, particularly adenocarcinomas, low IL-1β expression was associated with significantly improved OS. A similar pattern was observed in ALK-rearranged tumors. Our study indicated the distinct impact of IL-1β expression on survival outcomes across various NSCLC subgroups, which supports the potential utility of IL-1β as a biomarker for both prognosis and treatment stratification. Furthermore, they provide a rationale for future investigation of IL-1β as a predictive biomarker or therapeutic target, especially in combination with TKIs.

The disappointing results from the CANOPY trials may be partially explained by the choice of combination therapies. IL-1β blockade with chemotherapy or immunotherapy may not be the most effective modality in lung cancer treatment. Instead, a combinatorial approach involving targeted therapies against key oncogenic driver mutations such as combining EGFR TKIs with IL-1β is an area worth exploring. A retrospective study [36] involving 463,679 individuals showed that exposure to air pollution can trigger IL-1β release, promoting mutation-driven lung cancer development in never-smokers. The study also showed that IL-1β blockade could prevent particulate matter-induced tumor formation in mouse models with EGFR and KRAS mutations. Additional preclinical study demonstrated that IL-1β induces EH domain-containing protein 1 expression, promoting the epithelial-to-mesenchymal transition and EGFR TKI resistance [23]. Together, these data suggest that IL-1β blockade might be particularly effective in treating lung cancer in the presence of oncogenic driver mutations, potentially making canakinumab more effective when combined with EGFR TKIs rather than with chemotherapy or immunotherapy. However, no trials to date have investigated the efficacy of combining canakinumab with EGFR TKIs in NSCLC.

We also evaluated the impact of IL-1β on treatment duration, measured by TOT. In the overall group treated with pembrolizumab, IL-1β levels did not significantly affect TOT, indicating limited predictive value in unselected populations. However, in the KRAS-mutant adenocarcinoma subgroup, patients with high IL-1β expression had a modest but statistically significant increase in TOT compared to those with low expression, despite no difference in OS. This discrepancy may reflect the influence of subsequent lines of therapy or crossover, as OS captures outcomes beyond the first course of treatment. It is also possible that the modest TOT benefit reflects transient modulation of treatment sensitivity by IL-1β rather than a durable survival effect. The limited sample size within this subgroup may have further reduced power to detect OS differences. While the clinical relevance of this difference is limited, this contrast highlights the complexity of IL-1β’s effects on treatment response and disease trajectory. While IL-1β expression may not consistently predict TOT across all subgroups, it may still influence treatment outcomes in specific contexts. Future research should delve deeper into these relationships to develop more personalized treatment strategies based on IL-1β expression and other biomarkers, while emphasizing the complex role of IL-1β in NSCLC and its interaction with different treatments.

Further, the observed association between high expression of IL-1β and specific immune cells that infiltrated the TME suggests that IL-1β contributes to the complex interplay of immune cells. While several immune populations were more prevalent in IL-1β-high tumors, the degree of change varied, pointing to the complex role of IL-1β in modulating the immune landscape of NSCLC. These findings are consistent with previous data on the roles of IL-1β in shaping the TME, and they underscore the need to consider immune context when designing IL-1β-targeted therapies, as the immune context may influence the response to such treatments [37,38,39,40,41,42,43].

IL-1β expression was positively associated with TP53 mutations, TMB-high status, and PD-L1 expression, suggesting a link between IL-1β-driven inflammation and genomic instability in NSCLC. The association with TP53 mutations is consistent with evidence that TP53 loss enhances pro-inflammatory signaling, including IL-1β upregulation, which may contribute to tumor progression. Likewise, the correlation with PD-L1 expression suggests that IL-1β may promote an immunosuppressive TME by driving PD-L1 upregulation on tumor or immune cells, thereby facilitating immune evasion. Conversely, its negative association with EGFR and KRAS mutations suggests divergent roles across molecular subtypes, possibly due to the distinct pathways driving tumorigenesis in these contexts. These associations may be explained by known biological mechanisms. High IL-1β expression has been shown to enhance pro-inflammatory signaling in TP53-mutant tumors, promote PD-L1 upregulation and immune evasion, and support tumor progression through angiogenesis and recruitment of myeloid-derived suppressor cells and tumor-associated macrophages [15,29,44,45,46,47]. In contrast, tumors driven primarily by EGFR or KRAS mutations may rely on constitutive kinase pathway activation, reducing dependency on IL-1β–mediated signaling. Together, these findings indicate that IL-1β may act as both a biomarker of inflammation and a functional mediator of tumor behavior, particularly in genetically defined subgroups such as TP53-mutated or PD-L1-high NSCLC. Preliminary results from this study were presented at the 2024 ASCO Annual Meeting [48].

Our study has several limitations. Its retrospective design and the heterogeneity of the clinical data, including the absence of individual-level characteristics and comorbidities, limited our ability to perform multivariate analyses. The lack of staging information prevented stratified analysis by disease stage. While our large sample size improves the statistical power, it may also yield statistically significant findings that are not necessarily clinically meaningful. In addition, IL-1β expression was evaluated using bulk RNA sequencing, which cannot distinguish between tumor cells and other immune or stromal populations in the TME. Immune fractions were estimated using the quanTIseq algorithm, which provides computationally derived estimates based on reference gene signatures, but does not capture single-cell resolution. Moreover, we did not assess IL-1β protein levels or activity, and post-transcriptional regulation may lead to discordance between transcript and protein expression. These factors limit our ability to infer the functional impact of IL-1β, and future studies incorporating single-cell RNA sequencing, proteomic analyses, and functional assays will be important to validate and extend our findings. Taken together, these limitations warrant cautious interpretation of our findings.

Despite these challenges, this study provides strong rationale for further investigation of IL-1β in NSCLC. The findings suggest that IL-1β may serve as a biomarker of prognosis and potential treatment response, particularly in EGFR- and ALK-mutated tumors. Given the absence of trials testing IL-1β inhibitors in combination with TKIs, future studies should evaluate this strategy in genetically defined NSCLC subsets to determine whether dual targeting of oncogenic signaling and inflammation can improve outcomes.

## 5. Conclusions

This study demonstrates that IL-1β expression has meaningful prognostic significance in NSCLC, particularly in tumors harboring oncogenic driver mutations. Low IL-1β expression correlates with improved OS in EGFR- and ALK-altered tumors, suggesting that IL-1β expression can inform both prognosis and therapeutic strategies in NSCLC, including IL-1β-targeted approaches in precision medicine.

Overall, this study supports further exploration of IL-1β as a key modulator in NSCLC pathogenesis and treatment response. Future clinical trials should explore IL-1β inhibitors in combination with targeted therapies in molecularly defined subgroups, which have the potential to pave the way for more personalized and effective treatment strategies in lung cancer, ultimately improving patient outcomes.

## Figures and Tables

**Figure 1 cancers-17-02895-f001:**
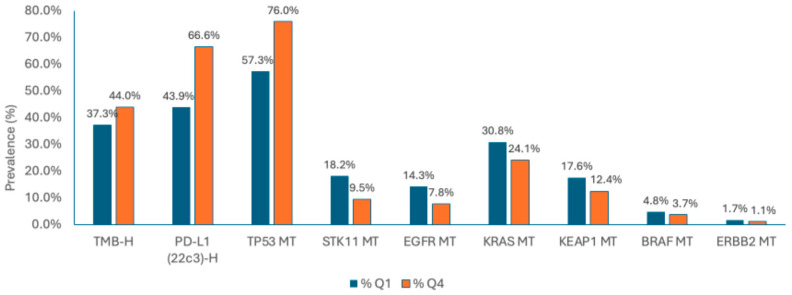
Prevalence of common genomic alterations in tumors with low (IL-1β Q1) and high (IL-1β Q4) IL-1β expression. Bar graph showing the percentage of samples with selected mutations stratified by IL-1β expression quartiles. All listed differences are statistically significant (*q* < 0.01). Abbreviations: BRAF, B-Raf proto-oncogene; EGFR, epidermal growth factor receptor; ERBB2, erb-b2 receptor tyrosine kinase 2; KEAP1, Kelch-like ECH-associated protein 1; KRAS, Kirsten rat sarcoma virus; MT, mutated; PD-L1, programmed death-ligand 1; STK11, serine/threonine kinase 11; TMB-H, tumor mutational burden high; TP53, tumor protein p53.

**Figure 2 cancers-17-02895-f002:**
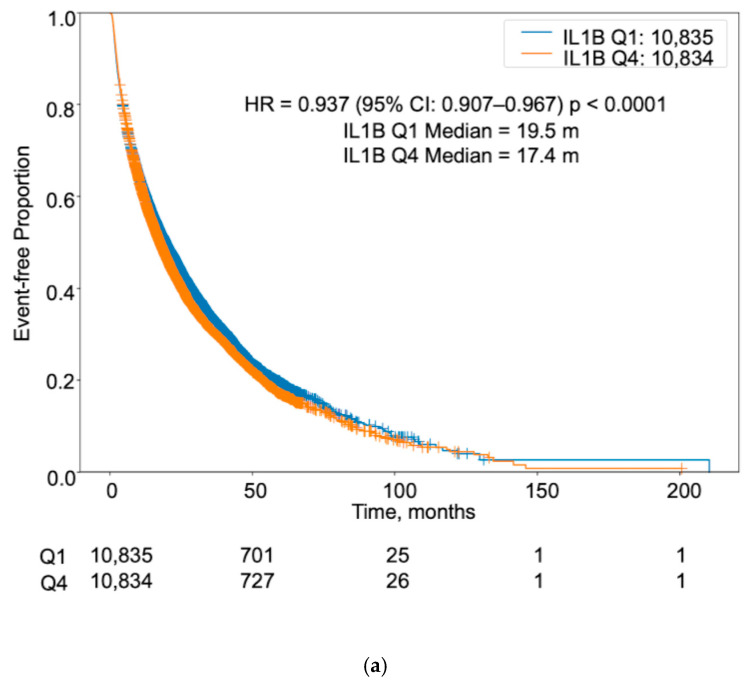
Overall survival in NSCLC by IL-1β expression and oncogenic driver subtype. Kaplan-Meier curves comparing overall survival (OS) between IL-1β Q1 (low expression) and Q4 (high expression) groups. (**a**) All patients with NSCLC. (**b**) Patients with EGFR-mutated NSCLC, regardless of histology. (**c**) Patients with EGFR-mutated adenocarcinoma. (**d**) Patients with EGFR-mutated squamous cell carcinoma. (**e**) Patients with ALK-mutated NSCLC, regardless of histology. (**f**) Patients with ALK-mutated adenocarcinoma. Hazard ratios (HRs), 95% confidence intervals (CIs), and median OSs in months are shown. Log-rank *p* values reflect differences in survival between Q1 and Q4 groups.

**Table 1 cancers-17-02895-t001:** Baseline Characteristics of the Study Population by IL-1β Expression Quartiles.

Characteristic	IL-1β Q1 (*N* = 10,849)	IL-1β Q4 (*N* = 10,849)	*p* Value
Sex			
Male	5365 (49.5%)	5670 (52.3%)	<0.001
Female	5484 (50.5%)	5179 (47.7%)	
Median Age at Specimen Collection	69 (12– > 89)	70 (21– > 89)	N.S.
Histology			
Adenocarcinoma	7620 (72.2%)	4803 (44.3%)	<0.001
Squamous cell carcinoma	1309 (12.1%)	3952 (36.4%)	<0.001
Neuroendocrine carcinoma	197 (1.8%)	117 (1.1%)	<0.001
Adenosquamous	75 (0.7%)	123 (1.1%)	0.001
Sarcomatoid	19 (0.2%)	54 (0.5%)	N.S.
Large Cell	28 (0.3%)	32 (0.3%)	N.S.
Other	1601 (14.8%)	1768 (16.3%)	N.S.
Primary vs. Metastatic			
Primary	6508 (60.0%)	7466 (68.6%)	<0.001
Metastatic	4071 (37.5%)	3066 (28.3%)	<0.001
Unknown	270 (2.5%)	317 (2.9%)	N.S.

Abbreviations: N, number; N.S., not significant.

**Table 2 cancers-17-02895-t002:** Median cell fraction of different types of immune cell in IL-1β Q1 and Q4 groups.

	Prevalence in IL-1β Q1	Prevalence in IL-1β Q4	*p* Value	*q* Value
Neutrophil	4.70%	7.00%	<0.001	<0.001
Macrophage M1	3.70%	5.90%	<0.001	<0.001
Macrophage M2	4.90%	5.00%	<0.001	<0.001
B cell	3.80%	4.50%	<0.001	<0.001
T cell regulatory	2.20%	2.90%	<0.001	<0.001
NK cell	2.40%	2.50%	<0.001	<0.001
T cell CD8+	0.40%	0.90%	<0.001	<0.001
Myeloid dendritic cell	0.30%	0.50%	<0.001	<0.001
Monocyte	0.00%	0.00%	NA	NA
T cell CD4+ (non-regulatory)	0.00%	0.00%	NA	NA

Abbreviations: NA, not applicable; NK, natural killer.

**Table 3 cancers-17-02895-t003:** Comparison of IL-1β Expression Between Adenocarcinomas and Squamous Cell Carcinomas by Molecular Subgroup.

	Median IL-1β Expression (TPM)				Adeno vs. Squamous	
	Adeno	N	Squamous	N	*p* Value	*q* Value
EGFR mutation	4.44	2983	9.98	75	<0.001	<0.001
KRAS mutation	4.78	6098	10.14	247	<0.001	<0.001
ALK fusion	5.16	551	4.88	9	N.S.	N.S.
EGFR or KRAS or ALK positive	4.67	9605	10.14	329	<0.001	<0.001
No EGFR/KRAS/ALK	4.92	11262	8.47	7915	<0.001	<0.001

Abbreviations: adeno, adenocarcinoma; squamous, squamous cell carcinoma; N, sample number; N.S., not significant; TPM, transcripts per million.

## Data Availability

The data sets analyzed during this study are not publicly available but are available from the corresponding author on reasonable request.

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
