# Peer review of "Prognostic Significance and Emerging Predictive Potential of Interleukin-1β Expression in Oncogene-Driven NSCLC"

_cancers, 2025, doi:10.3390/cancers17172895_

Round 1
Reviewer 1 Report
Comments and Suggestions for Authors
The authors examined a large number of tumor samples to study whether the level of interleukin-1ß correlates with the overall survival of non-small cell lung cancer (NSCLC) patients. Especially, investigating this correlation in cancers with specific oncogenic mutations makes this study noteworthy and of interest to the readers. The study is well designed, and the statistical data analysis was performed correctly.
My major concern of the study is that tissue analysis is limited to DNA and RNA sequencing. Single cell RNA sequencing and proteomics data could have further supported the study and given better insights into the stated question. Interleukin-1ß can be expressed by different cell populations and the responses by different cell types can vary. Therefore, a single cell RNAseq is more appropriate.
There are multiple mechanisms of post-transcriptional regulations within the cell such as through non-coding RNAs which could affect the protein level. This study lacks the information whether the interleukin-1ß transcript level correlates with the protein level.
However, I do acknowledge the large sample size and the efforts of the authors to estimate the immune cells in tumor microenvironment by data deconvolution. I suggest the authors to add the above-mentioned limitations in their discussion section.
Here are two minor concerns:
- Please check the sample number. Abstract “We analyzed 21,669 NSCLC tumors”, line 31. Materials and Methods: “21,698 formalin-fixed, paraffin-embedded (FFPE) tumor samples”, line 84. See line 148 as well.
- All figure labels are too small (at least in the format I received). Please increase the font size.
Author Response
The authors examined a large number of tumor samples to study whether the level of interleukin-1ß correlates with the overall survival of non-small cell lung cancer (NSCLC) patients. Especially, investigating this correlation in cancers with specific oncogenic mutations makes this study noteworthy and of interest to the readers. The study is well designed, and the statistical data analysis was performed correctly.
My major concern of the study is that tissue analysis is limited to DNA and RNA sequencing. Single cell RNA sequencing and proteomics data could have further supported the study and given better insights into the stated question. Interleukin-1ß can be expressed by different cell populations and the responses by different cell types can vary. Therefore, a single cell RNAseq is more appropriate.
There are multiple mechanisms of post-transcriptional regulations within the cell such as through non-coding RNAs which could affect the protein level. This study lacks the information whether the interleukin-1ß transcript level correlates with the protein level.
However, I do acknowledge the large sample size and the efforts of the authors to estimate the immune cells in tumor microenvironment by data deconvolution. I suggest the authors to add the above-mentioned limitations in their discussion section.
-We thank the reviewer for this insightful comment! We agree that our study is limited by the use of bulk DNA/RNA sequencing, which cannot distinguish between different cell populations contributing to IL-1β expression. Furthermore, we did not include proteomic analyses, and therefore cannot confirm whether IL-1β transcript levels correspond to protein levels within the TME. We have now added these limitations to our discussion section.
Here are two minor concerns:
1. Please check the sample number. Abstract “We analyzed 21,669 NSCLC tumors”, line 31. Materials and Methods: “21,698 formalin-fixed, paraffin-embedded (FFPE) tumor samples”, line 84. See line 148 as well.
-We thank the reviewer for noting the discrepancy in sample numbers. We have corrected the Abstract to report 21,698 tumors, consistent with the Methods and Results.
2. All figure labels are too small (at least in the format I received). Please increase the font size.
-We thank the reviewer for this helpful observation. We have revised all figures to increase the font size to ensure readability in the published format.

Reviewer 2 Report
Comments and Suggestions for Authors
The paper evaluated the prognostic and predictive significance of IL-1β expression across NSCLC subtypes, which may support the research of IL-1β-targeted strategies in EGFR- or ALK-altered tumors. The paper is well designed. while there are some problems to be discussed.
1.The study did not clearly explain how the cut-off value for IL-1β expression level was determined (such as the dividing point between Q1 and Q4). It is suggested to provide supplementary explanations on the stratification criteria for IL-1β expression, and whether they are based on previous studies or internal validation data.
2.In the KRAS mutant subgroup, high IL-1β expression was associated with a longer time to treatment (TOT), but there was no difference in overall survival (OS). The clinical significance of this result is unclear. It is suggested to discuss the possible reasons for the inconsistency between TOT and OS, such as whether it is related to subsequent treatment or insufficient sample size.
3.Figure 1 shows the correlation between IL-1β and various genetic mutations, but does not discuss the potential mechanisms underlying these associations. Recommendation: In the discussion section, supplement possible biological mechanisms, such as the interaction between IL-1β and TP53 mutations or PD-L1 expression.
4.Some sentences are lengthy or not clear enough (such as the last sentence of the abstract). It is recommended to simplify the language to ensure logical coherence. For example, change "These findings support exploration of IL-1β-targeted strategies, particularly in EGFR- or ALK-altered tumors" to "These findings suggest that IL-1β-targeted strategies may be particularly relevant in EGFR- or ALK-altered tumors."
5.Abbreviations (such as TOT, TME) were not defined when they first appeared. It is recommended to provide the full name when they are first introduced, for example, "time on treatment (TOT)".
Author Response
1.The study did not clearly explain how the cut-off value for IL-1β expression level was determined (such as the dividing point between Q1 and Q4). It is suggested to provide supplementary explanations on the stratification criteria for IL-1β expression, and whether they are based on previous studies or internal validation data.
-We thank the reviewer for this comment. We have clarified in the Methods how IL-1β cut-offs were determined. Quartile cutoffs (Q1 vs Q4) were derived from the distribution of expression values across the cohort. We also tested a median (50%) cut-off, but this did not yield significant differences in survival outcomes (data not shown). Therefore, quartile stratification was selected as the primary analytic approach.
2.In the KRAS mutant subgroup, high IL-1β expression was associated with a longer time to treatment (TOT), but there was no difference in overall survival (OS). The clinical significance of this result is unclear. It is suggested to discuss the possible reasons for the inconsistency between TOT and OS, such as whether it is related to subsequent treatment or insufficient sample size.
-We thank the reviewer for raising this important point. We have added text to the Discussion noting that the discrepancy between TOT and OS in the KRAS-mutant subgroup may reflect the impact of subsequent therapies, transient treatment effects of IL-1β, as well as limited sample size.
3.Figure 1 shows the correlation between IL-1β and various genetic mutations, but does not discuss the potential mechanisms underlying these associations. Recommendation: In the discussion section, supplement possible biological mechanisms, such as the interaction between IL-1β and TP53 mutations or PD-L1 expression.
-We appreciate this insightful suggestion. We have revised the Discussion section to describe potential biological mechanisms underlying the observed associations, including the role of TP53 loss in IL-1β upregulation and the possibility that IL-1β signaling contributes to PD-L1 expression and immune evasion.
4.Some sentences are lengthy or not clear enough (such as the last sentence of the abstract). It is recommended to simplify the language to ensure logical coherence. For example, change "These findings support exploration of IL-1β-targeted strategies, particularly in EGFR- or ALK-altered tumors" to "These findings suggest that IL-1β-targeted strategies may be particularly relevant in EGFR- or ALK-altered tumors."
-We thank the reviewer for this suggestion. We have revised the final sentence of the Abstract for clarity and now state: “These findings suggest that IL-1β-targeted strategies may be particularly relevant in EGFR- or ALK-altered tumors.”
5.Abbreviations (such as TOT, TME) were not defined when they first appeared. It is recommended to provide the full name when they are first introduced, for example, "time on treatment (TOT)".
-We appreciate the reviewer’s observation. In the revised manuscript, we have ensured that all abbreviations are defined at first appearance in the main text. Specifically, time on treatment (TOT) is now defined in the Methods section at its first use, and tumor microenvironment (TME) is defined in the Introduction at first mention. We also reviewed the full text to confirm consistency of abbreviation usage throughout.

Reviewer 3 Report
Comments and Suggestions for Authors
The manuscript entitled “Prognostic Significance and Emerging Predictive Potential of Interleukin-1β Expression in Oncogene-Driven NSCLC” intends to analyze the potential of IL-1β as a diagnostic and therapeutic marker of oncogene-driven NSCLC of the clinical samples. The authors performed gene sequencing and IHC of the specimens and attempted to demonstrated the pro-inflammatory marker IL-1β correlation with certain subtypes of oncogene-driven NSCLC samples. Overall, the manuscript has adequately demonstrated the potential of IL-1β as a therapeutic and predictive tool for some subtypes of mutation-driven NSCLC. The manuscript is well written, a minor grammatical error in line 124-125, that we could identify. The manuscript may be recommended for publication with minor changes.
Author Response
The manuscript entitled “Prognostic Significance and Emerging Predictive Potential of Interleukin-1β Expression in Oncogene-Driven NSCLC” intends to analyze the potential of IL-1β as a diagnostic and therapeutic marker of oncogene-driven NSCLC of the clinical samples. The authors performed gene sequencing and IHC of the specimens and attempted to demonstrated the pro-inflammatory marker IL-1β correlation with certain subtypes of oncogene-driven NSCLC samples. Overall, the manuscript has adequately demonstrated the potential of IL-1β as a therapeutic and predictive tool for some subtypes of mutation-driven NSCLC. The manuscript is well written, a minor grammatical error in line 124-125, that we could identify. The manuscript may be recommended for publication with minor changes.
-We thank the reviewer for this observation. The sentence in question has since been revised during the current round of edits to incorporate additional details about the deconvolution method (as suggested by another reviewer). The revised version now reads:
“Immune cell fractions in the TME were estimated using the quanTIseq deconvolution algorithm, which infers immune composition from bulk RNA-seq data based on validated cell-type-specific transcriptomic signatures.”
We believe this addresses both the grammar concern and ensures greater clarity.

Reviewer 4 Report
Comments and Suggestions for Authors
The authors examined the association between disease features and IL-1b expression levels in over 20,000 NSCLC patients, and concluded that IL-1b can be a prognostic markers for NSCLC, as lower level of IL-1b is associated with better survival. The study is interesting and of potential significance, but could be further improved as suggested below:
- The authors mentioned the role of IL-1b in NSCLC. a more detailed discussion/introduction about cells that produce IL-1b, its target cells, its effector mechanisms and what are the roles of IL-1b in other cancers could provide more perspectives.
- The authors showed the freqeuncy of immune cell lineages in different patient samples without mentioning how such lineages were identified (i.e markers and signature genes), and what single cell data set is used to deconvolve the bulk RNA seq data.
- Potential mechanisms/pathways behind the association discovered by the authors should be discussed.
Author Response
The authors examined the association between disease features and IL-1b expression levels in over 20,000 NSCLC patients, and concluded that IL-1b can be a prognostic markers for NSCLC, as lower level of IL-1b is associated with better survival. The study is interesting and of potential significance, but could be further improved as suggested below:
- The authors mentioned the role of IL-1b in NSCLC. a more detailed discussion/introduction about cells that produce IL-1b, its target cells, its effector mechanisms and what are the roles of IL-1b in other cancers could provide more perspectives.
-We thank the reviewer for this suggestion. In the revised Introduction, we have expanded the description of IL-1β biology. We now note its major cellular sources (monocytes, macrophages, neutrophils, and tumor/stromal cells), its target populations within the tumor microenvironment (endothelial cells, fibroblasts, immune subsets), and its key effector mechanisms (angiogenesis, myeloid recruitment, chronic inflammation). We also highlight evidence from other malignancies, including breast, gastric, and colorectal cancers.
- The authors showed the frequency of immune cell lineages in different patient samples without mentioning how such lineages were identified (i.e markers and signature genes), and what single cell data set is used to deconvolve the bulk RNA seq data.
- We thank the reviewer for this important comment. We have clarified in the Methods that immune cell fractions were estimated using the quanTIseq deconvolution algorithm (Finotello et al., Genome Med. 2019), which infers immune composition from bulk RNA-seq data based on validated transcriptomic signatures. We have also expanded the Discussion to acknowledge the limitations of bulk RNA-seq and computational deconvolution compared with single-cell RNA-seq.
- Potential mechanisms/pathways behind the association discovered by the authors should be discussed.
-We thank the reviewer for this helpful suggestion. In the revised Discussion, we have added a section describing potential mechanisms that may explain the observed associations between IL-1β expression, oncogenic drivers, and immune features. Specifically, we note that IL-1β can promote genomic instability via TP53 dysregulation, drive PD-L1 upregulation and immune evasion, and enhance tumor progression through angiogenesis and recruitment of immunosuppressive myeloid populations. Conversely, its inverse association with EGFR and KRAS mutations may reflect divergent oncogenic signaling pathways.
